

# Upregulation of HPV16E1 and E7 expression and FOXO3a mRNA downregulation in high-grade cervical neoplasia

Thanayod Sasivimolrattana[1,2], Aileen Gunawan[3], Warattaya Wattanathavorn[2], Chavis Pholpong[2], Arkom Chaiwongkot[2,4], Pattarasinee Bhattarakosol[5] and Parvapan Bhattarakosol[2,4]

[1] Department of Microbiology, Faculty of Public Health, Mahidol University, Bangkok, Thailand
[2] Center of Excellence in Applied Medical Virology, Department of Microbiology, Faculty of Medicine, Chulalongkorn University, Bangkok, Thailand
[3] Department of Biomedicine, School of Life Sciences, Indonesia International Institute for Life Sciences, Jakarta, Indonesia
[4] Division of Virology, Department of Microbiology, Faculty of Medicine, Chulalongkorn University, Bangkok, Thailand
[5] Department of Mathematics and Computer Science, Faculty of Science, Chulalongkorn University, Bangkok, Thailand

Corresponding author
Parvapan Bhattarakosol, parvapan.b@chula.ac.th

## ABSTRACT

**Background**. Cervical cancer remains a significant global health concern, ranking as the fourth most prevalent cancer among women worldwide. Human papillomaviruses (HPV) transcribe many genes that might be responsible for cervical cancer development. This study aims to investigate the correlation between the expression of HPV16 early genes and the mRNA expression of human FOXO3a, a tumor suppressor gene, in association with various stages of cervical precancerous lesions.

**Methods**. Eighty-five positive HPV16 DNA cervical swab samples were recruited and categorized based on cytology stages, *i.e.*, negative for intraepithelial lesion or malignancy (NILM), atypical squamous cells of undetermined significance (ASC-US), low-grade squamous intraepithelial lesion (LSIL), atypical squamous cell cannot exclude HSIL (ASC-H), high-grade squamous intraepithelial lesion (HSIL). RT-qPCR was performed to amplify HPV16E1, E4, E6, E6*I, E7, and human FOXO3a mRNA expression in all samples. The relative expression of those genes was calculated using GAPDH as a control. Detection of FOXO3a mRNA expression in the cervical cancer cell line by RT-qPCR and meta-analysis of FOXO3a expression using the RNA-Seq dataset by GEPIA2 were analyzed to support the conclusions.

**Results**. Among the cervical samples, HPV16E1 and E7 were significantly increased expression correlating to disease severity. HPV16E4 mRNA expression was 100% detected in all LSIL samples, with a significant increase observed from normal to LSIL stages. Conversely, FOXO3a mRNA expression decreased with disease severity, and the lowest expression was observed in HSIL/squamous cell carcinoma (SCC) samples. In addition, similar results of FOXO3a downregulation were also found in the cervical cancer cell line and RNA-Seq dataset of cervical cancer samples.

**Conclusion**. HPV16 early mRNA levels, including E1 and E7, increase during cancer progression, and downregulation of FOXO3a mRNA is a characteristic of cervical

cancer cells and HSIL/SCC. Additionally, HPV16E4 mRNA expression was consistently detected in all LSIL samples, suggesting the presence of active viral replication. These findings might lead to further investigation into the interplay between HPV gene expression and host cell factors for targeted therapeutic strategies in cervical cancer management.

## INTRODUCTION

Cervical cancer (CC) ranks as the fourth most prevalent cancer among women across the globe. In 2018, there were approximately 600,000 new cases of CC reported worldwide, leading to a significant mortality rate of around 300,000 deaths annually (*Cohen et al., 2019*). About 70% of cases of cervical cancer are caused by the human papillomavirus (HPV) type 16 (HPV16), which is classified as a high-risk HPV (hrHPV) (*Cerasuolo et al., 2017*). The American Cancer Society Guidelines recommend routine CC screening for individuals who are at least 25 years of age (*Fontham et al., 2020*). The primary technique is the Papanicolaou test, also known as the Pap test or Pap smear. The purpose of the test is to identify potentially precancerous changes. Cytology results are categorized as negative for intraepithelial lesion or malignancy (NILM), atypical squamous cells of undetermined significance (ASC-US), low-grade squamous intraepithelial lesion (LSIL), atypical squamous cell cannot exclude HSIL (ASC-H), high-grade squamous intraepithelial lesion (HSIL), and squamous cell carcinoma (SCC) (*Lowy & Schiller, 2006*).

HPV16 is a double-stranded DNA virus without an envelope, mainly infecting mucosal and epithelial cells. The viral genome consists of various early (E) genes, *i.e.,* E1, E2, E4, E5, E6, E7, and late (L) genes, *i.e.,* L1, and L2 genes (*Yu, Majerciak & Zheng, 2022*). E genes are responsible for the viral life cycle, whereas L genes are viral structural proteins (*Bhattacharjee et al., 2022*). E1 plays a role in initiating and regulating HPV replication through its helicase function, while E2 collaborates with E1 to initiate viral replication (*Bhattacharjee et al., 2022*). E4 is involved in viral release, transmission, and post-translational modification (*Bhattacharjee et al., 2022*). Among the E genes of hrHPVs, at least three genes, *i.e.,* E5, E6, and E7, are recognized as oncogenes. E6 and E7 trigger cell cycle entry, thereby stimulating DNA synthesis and preventing apoptosis in infected cells (*Hawley-Nelson et al., 1989*). E6 degrades the p53 tumor suppressor protein, while E7 controls cell proliferation by interacting with the retinoblastoma protein (pRB) and p107 (*Egawa & Doorbar, 2017*). Beside a full-length E6, truncated E6 proteins, commonly named E6*, were found during the splicing of polycistronic early transcripts. Spliced E6*I transcripts are the most abundant RNAs produced in HPV-related cancers (*Paget-Bailly et al., 2019*). E5 is attributed to its ability to prompt anchorage-independent growth, more efficient growth in low serum, and tumorigenic transformation of murine keratinocytes and fibroblasts, as well as stimulate cellular DNA synthesis in human keratinocytes (*Pim, Collins & Banks, 1992*).

Recently, our group proposed the possibility of HPV16E1 as another potential oncogene (*Baedyananda et al., 2022*). We found that HPV16E1 significantly increased expression in cancer stages (*Baedyananda, Chaiwongkot & Bhattarakosol, 2017*). In addition, HPV16E1-downregulated cervical cancer cells altered the functions of cell viability, colony formation, invasion, and anchorage-independent cell growth (*Sasivimolrattana, Chaiwongkot & Bhattarakosol, 2023*). This also impacted the cellular FOXO3a signaling pathway (*Sasivimolrattana, Chaiwongkot & Bhattarakosol, 2023*). FOXO3a functions as a tumor suppressor by regulating genes involved in apoptosis, cell cycle arrest, oxidative stress resistance and autophagy (*Nho & Hergert, 2014*). Interestingly, the correlation of FOXO3a expression and cancer progression has been shown in several kinds of tumors. In gastric adenocarcinoma, lower FOXO3a levels correlate with larger tumor size, poor histopathological classification, and worse overall survival (*Yang et al., 2013*). Decreased FOXO3a expression correlates with increased cancer cell progression, promoting tumor occurrence, metastasis, and aggressive features in various malignancies, including upper tract urothelial carcinoma (*Zhang et al., 2021*). However, no evidence in clinical specimen supports HPV16E1 and FOXO3a expression in association with disease progression. Therefore, this study aims to uncover the correlation between E1 expression as well as other early genes (E4, E6, E6*I, and E7) and the mRNA expression of FOXO3a in different stages of cervical precancerous lesions.

# MATERIALS & METHODS

## Ethical approval

This study was approved by the Institutional Review Board (IRB) of the faculty of Medicine, Chulalongkorn University, a WHO certified ethics committee (COA No.0051/2023, IRB No.0035/66). The protocol of the study was also approved by Institutional Biosafety Committee (IBC) of the Faculty of Medicine, Chulalongkorn University (MDCU-IBC006/2023).

## Clinical samples

Leftover 85 HPV16 DNA-positive cervical swab samples collected in the ThinPrep® Pap Test (Hologic, Marlborough, MA, USA) were selected from all cervical swab samples sent for routine HPV screening using the Cobas 4800 high-risk HPV test (Roche, Basel, Switzerland) at the Virology Unit, Department of Microbiology, King Chulalongkorn Memorial Hospital, Bangkok, Thailand. These samples were cytologically diagnosed prior to being sent to perform HPV screening test. They were categorized based on cytology stages, including NILM, ASC-US, LSIL, ASC-H, and HSIL/SCC. The exclusion criteria were any samples obtained from women with active microbial infections and those who were menstruating on the day of specimen collection. The selected samples in ThinPrep were stored at 4 °C before RNA isolation. The RNA extraction and cDNA synthesis were performed within one month after sample collection.

In this study, leftover cervical swab samples were obtained from the Virology laboratory, Microbiology Department, King Chulalongkorn Memorial Hospital, Thai Red Cross, Bangkok, Thailand. In accordance with IRB regulations, the hospital director's permission

was given without the patient's consent, which is a prerequisite for IRB approval. A total of 85 leftover cervical swab samples that tested positive for HPV16 were enrolled in this study.

## Real-time PCR

Total RNA of each cervical sample was extracted by the RNeasy® Mini Kit (QIAGEN, USA) according to the manufacturer's protocol. The yield and purity of the isolated RNA were assessed through Nanodrop (Eppendorf, Hamburg, Germany). First-strand cDNA synthesis was constructed by SuperScript™ III Reverse Transcriptase (Invitrogen, Waltham, MA, USA) according to the manufacturer's protocol using 5 µL of RNA input. HPVE1, E4, E6, E6*I, E7, and FOXO3a mRNA expression were amplified by SsoAdvanced™ Universal SYBR® Green Supermix (BioRad, Hercules, CA, USA) according to the manufacturer's protocol through QuantStudio™ 5 Real-Time PCR System (Applied Biosystems, La Jolla, CA, USA) using 5 µL of cDNA input. The selected primers in this study were shown in Table S1 (*Baedyananda, Chaiwongkot & Bhattarakosol, 2017*; *Bogovac et al., 2011*; *Chaiwongkot et al., 2020*; *Joseph, Srivastava & Pfister, 2012*; *Kannike et al., 2014*; *Supchokpul et al., 2011*). The RT-qPCR conditions included polymerase activation and DNA denaturation at 95 °C for 30 s, and 40 cycles of amplification, which included denaturation at 95 °C for 15 s and annealing/extension either at 63 °C for 30 s (for HPVE4) or 60 °C for 60 s (for other targets). GAPDH was used as a housekeeping gene. 10 µM of each primer was used in each reaction. The fold-change (ddCt) was determined by utilizing the Ct value derived from qPCR. The average Ct value of the NILM samples served as the control group. The specificity of the amplification was assessed through melt curve analysis. The data from each run will not be reliable if there is a Ct value from no template control (NTC). Moreover, the specificity of the primers for HPV early gene amplification was tested by using RNA from C-33A cells as a template. The amplification plot of those genes was undetectable, suggesting that the primers in this study are specific.

## Cell culture

SiHa cell (EP-CL-0210), an HPV16-positive (1–2 copies) human cervical carcinoma cell line isolated from fragments of a primary uterine tissue sample from a 55-year-old female Japanese patient with squamous cell carcinoma, was purchased from Elabscience Biotechnology (Houston, TX, USA). These cells were cultured in growth medium (Modified Eagle Medium (MEM; Gibco, Waltham, MA, USA)). CaSki cell (EP-CL-0048), an HPV16-positive human cervical carcinoma cell line containing approximately 600 copies of the HPV16 genome, was purchased from Elabscience Biotechnology. These cells were cultured in growth medium RPMI 1640 (Gibco). HeLa cell (CCL-2), an HPV18-positive human cervical carcinoma cell line, was kindly provided by Assoc. Prof. Siwaporn Boonyasuppayakorn, M.D., Ph.D., Faculty of Medicine, Chulalongkorn University, Thailand. This cell line was cultured in ready-to-use Dulbecco's Modified Eagle Medium (DMEM; Cat no. SH30022.02; Hyclone, Logan, UT, USA).

C-33 A cell, a human cervical cancer cell, isolated from the cervix of a 66-year-old, white, uterine cancer patient without HPV infection, was cultured in ready to use Dulbecco's Modified Eagle Medium.

All of the growth media were supplemented with 10% fetal bovine serum (FBS; Gibco), 0.01 M HEPES, 0.2% $NaHCO_3$, 100 units Penicillin, 100 $\mu$g streptomycin and sub-cultured every 3–4 days. The cells were grown under 5% $CO_2$ at 37 °C.

Primary Cervical Epithelial Cells (HCxECs; ATCC PCS-480-011) were purchased from the American Type Culture Collection (Manassas, VA, USA). These cells were sub-cultured with Trypsin-EDTA for Primary Cells (ATCC PCS-999-003) and the Trypsin Neutralizing Solution (ATCC PCS-999-004). Cervical Epithelial Cell Basal Medium (ATCC PCS-480-032) was used as a complete growth medium. The cells were grown under 5% $CO_2$ at 37 °C.

## GEPIA2

The online platform Gene Expression Profiling Interactive Analysis (GEPIA2; http://gepia2.cancer-pku.cn/) (accessed on 5 October 2024) (*Tang et al., 2019*) was used to compare the expression of FOXO3a between each cancer and the normal tissue. The datasets used for the analyses were obtained from The Cancer Genome Atlas (TCGA) and Genotype-Tissue Expression (GTEx) *via* the GEPIA2 web interface.

To observe the correlation between the expression of FOXO3a and the target of HPV16E6/7 oncoprotein, the RNA-seq data set of TCGA tumor (cervical squamous cell carcinoma and endocervical adenocarcinoma (CESC)) and TCGA normal was analyzed *via* the GEPIA2 web interface. Ensembl ID of FOXO3a, p53, pRB, and p107 were ENSG00000118689.14, ENSG00000141510.15, ENSG00000139687.13, and ENSG00000080839.11, respectively. Spearman's correlation was used to calculate the correlation between the genes.

## Statistical analysis

To observe the expression of each gene, the data was analyzed utilizing IBM® SPSS® Statistics 28 (IBM Corp., Armonk, NY, USA). The data (fold change) underwent transformation into log10 values to conduct normality tests and assess homogeneity. One-way analysis of variance (ANOVA) with Fisher's Least Square Difference (LSD) was employed to examine multiple comparisons among the normalized fold-change data across all cytology stages. The data was represented as mean log10 ± standard error of mean (SEM). Spearman's correlation was used for determining the correlation. Statistical significance (*) was indicated when *p*-value below 0.05. The box plot, bar chart, and graph were visualized by GraphPad Prism version 7 (GraphPad Software, Boston, MA, USA).

# RESULTS

## Characteristics of clinical samples

A total of 85 cervical swab samples positive for HPV16 were recruited. They were categorized according to the cytology characteristics. Among the cytology stages, NILM was the most common (45.88%), followed by ASC-US (29.41%), LSIL (9.41%), ASC-H (5.88%), and HSIL/SCC (9.41%), as shown in Table 1. Among those groups, the mean age distribution, ranging between 40.62 and 43.57 years, showed no statistically significant difference ($p = 0.6324$, one-way ANOVA) (Table 1).

**Table 1  Demographic data of the samples.**

| Cytology stage | Number | Age (years) | | |
| --- | --- | --- | --- | --- |
| | | Mean ± SEM | Range | Median |
| NILM | 39 | 43.92 ± 2.31 | 26–89 | 42.0 |
| ASC-US | 25 | 38.68 ± 2.25 | 24–71 | 37.0 |
| LSIL | 8 | 40.88 ± 4.31 | 28–57 | 36.5 |
| ASC-H | 5 | 43.60 ± 2.48 | 35–49 | 43.0 |
| HSIL/SCC | 8 | 44.13 ± 7.29 | 21–88 | 38.5 |

**Table 2  Number of HPV16E1, E4, E6, E6*I, E7, and FOXO3a mRNA expression (Log10-Fold Change) among different cytology stages.**

| Gene | Mean Log10 ± SEM of Fold change expression (Number positive sample; %) | | | | |
| --- | --- | --- | --- | --- | --- |
| | NILM $(N = 39)$ | ASC-US $(N = 25)$ | LSIL $(N = 8)$ | ASC-H $(N = 5)$ | HSIL/SCC $(N = 8)$ |
| HPV16E1 | 0.07 ± 0.19 (35; 89.74) | 0.64 ± 0.24 (23; 92) | 1.32 ± 0.70* (8; 100) | 0.88 ± 0.47 (4; 80) | 1.31 ± 0.31* (8; 100) |
| HPV16E4 | −0.08 ± 0.18 (21; 53.85) | 0.83 ± 0.29* (20; 80) | 1.08 ± 0.71* (8; 100) | 0.87 ± 0.39 (3; 60) | 0.89 ± 0.45 (7; 87.5) |
| HPV16E6 | −0.73 ± 0.32 (31; 79.49) | −0.20 ± 0.33 (24; 96) | −0.48 ± 0.60 (8; 100) | −0.15 ± 0.99 (4; 80) | 0.58 ± 0.58 (8; 100) |
| HPV16E6*I | 0.32 ± 0.39 (28; 71.79) | −0.28 ± 0.26 (21; 84) | 0.34 ± 0.66 (7; 87.5) | 0.59 ± 0.60 (4; 80) | −0.30 ± 0.38 (7; 87.5) |
| HPV16E7 | −0.29 ± 0.22 (35; 89.74) | 0.49 ± 0.28* (25; 100) | 1.06 ± 0.70* (8; 100) | 0.48 ± 0.47 (4; 80) | 0.91 ± 0.34* (8; 100) |
| FOXO3a | 0.18 ± 0.07# (35; 89.74) | 0.09 ± 0.11# (23; 92) | 0.14 ± 0.18# (6; 75) | 0.14 ± 0.07# (4; 80) | −0.45 ± 0.22 (8; 100) |

Notes.

*$p < 0.05$ when compared to NILM group

#$p < 0.05$ when compared to HSIL/SCC group

## HPV16 early gene expression in clinical specimens

The results indicated that HPV16 early genes (E1, E4, E6, E6*I, and E7) were expressed in all cytology stages (Table 2). Among those early genes, E4 had the least expression, especially in the NILM and ASC-H stages (53.85% and 60%, respectively). When the mRNA expression was assessed through fold change analysis, the expression of HPV16E1 and E7 were shown to be up-regulated related to the disease's severity (Table 2, Figs. 1A and 1E). A slight increase in HPV16E6 expression was observed without significance (Table 2, Fig. 1C). However, there was no change in HPV16E6*I expression among the groups (Table 2, Fig. 1D). Interestingly, a significant increase in HPV16E4 mRNA expression was observed in the lower stage of precancerous lesion groups (ASC-US and LSIL) compared to the control group (NILM) (Table 2, Fig. 1B). No significant difference in age was observed among the cytology stages in each of the E1, E4, E6, E6*I, and E7-positive samples (Table S2).

The correlation among each HPV early transcript expression was also observed. The expression level of HPV16E1 was well correlated with other HPV transcripts, especially HPV16E7 expression ($R = 0.8816$; $p < 0.001$) (Fig. 2A). Similar results were found between other HPV transcripts (Fig. 2).

## FOXO3a expression in clinical specimens

Human FOXO3a mRNA was also expressed in all cytology stages (Table 2). In contrast to HPV gene expression, human FOXO3a mRNA expression was statistically significantly

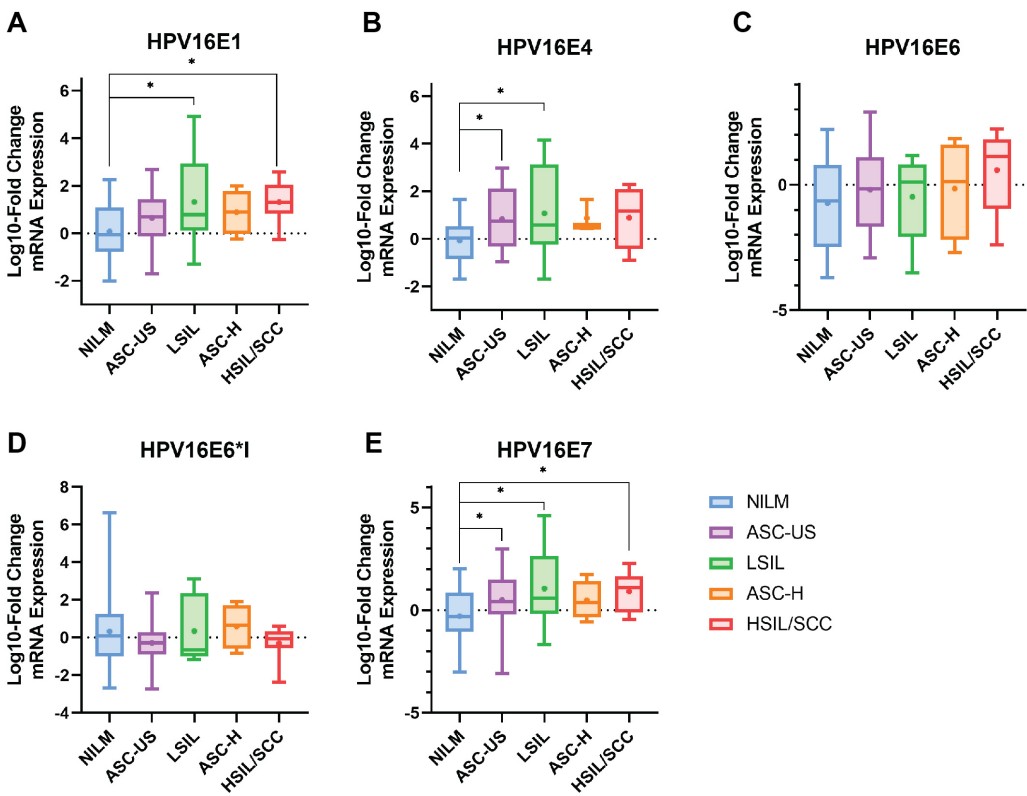

**Figure 1 The fold-change expression of HPV16's early transcript in different cytology stages.** The box plot showed median (line) and mean (dot) of log10-fold change mRNA expression. Statistical significance (*) was indicated for *p*-value below 0.05 (one-way ANOVA with Fisher's Least Square Difference). (A) HPV16E1, (B) HPVE4, (C) HPV16E6, (D) HPV16E6*I, (E) HPV16E7.

declined by the disease's severity (Table 2, Fig. 3A). Noted that the data of FOXO3a expression in ASC-US was not normally distributed. Interestingly, the lowest expression of FOXO3a was found in the HSIL/SCC group. To validate that FOXO3a was downregulated in the higher stages of cancer progression, cervical cancer cell lines with HPV DNA (SiHa, CaSki, and HeLa cells) and without HPV DNA (C-33A) were included, whereas the primary cervical epithelial cells (HCxECs) was used as the control. FOXO3a mRNA expression in the HCxECs (mean ± SEM = 2.092 ± 0.686) was significantly higher than those in all cervical cancer cell lines: C-33A (0.094 ± 0.012, $p = 0.0068$), CaSki (0.100 ± 0.011, $p = 0.0069$), SiHa (0.022 ± 0.004, $p = 0.0053$), and HeLa cells (0.031 ± 0.002, $p = 0.0055$) (Fig. 3B). No significantly different expression of FOXO3a among all cervical cancer cells was found. Moreover, according to the low number of high-grade cervical samples in our studies, GEPIA2 was utilized to analyze the RNA-seq data of pan-cancers from TCGA and the GTEx projects to confirm the impact of FOXO3a mRNA expression in the cervical cancers. The expression level of FOXO3a was actually low in cervical squamous cell carcinoma and endocervical adenocarcinoma (CESC) when compared to those in the matched normal tissues. In addition, downregulation of FOXO3a in lung adenocarcinoma (LUAD),

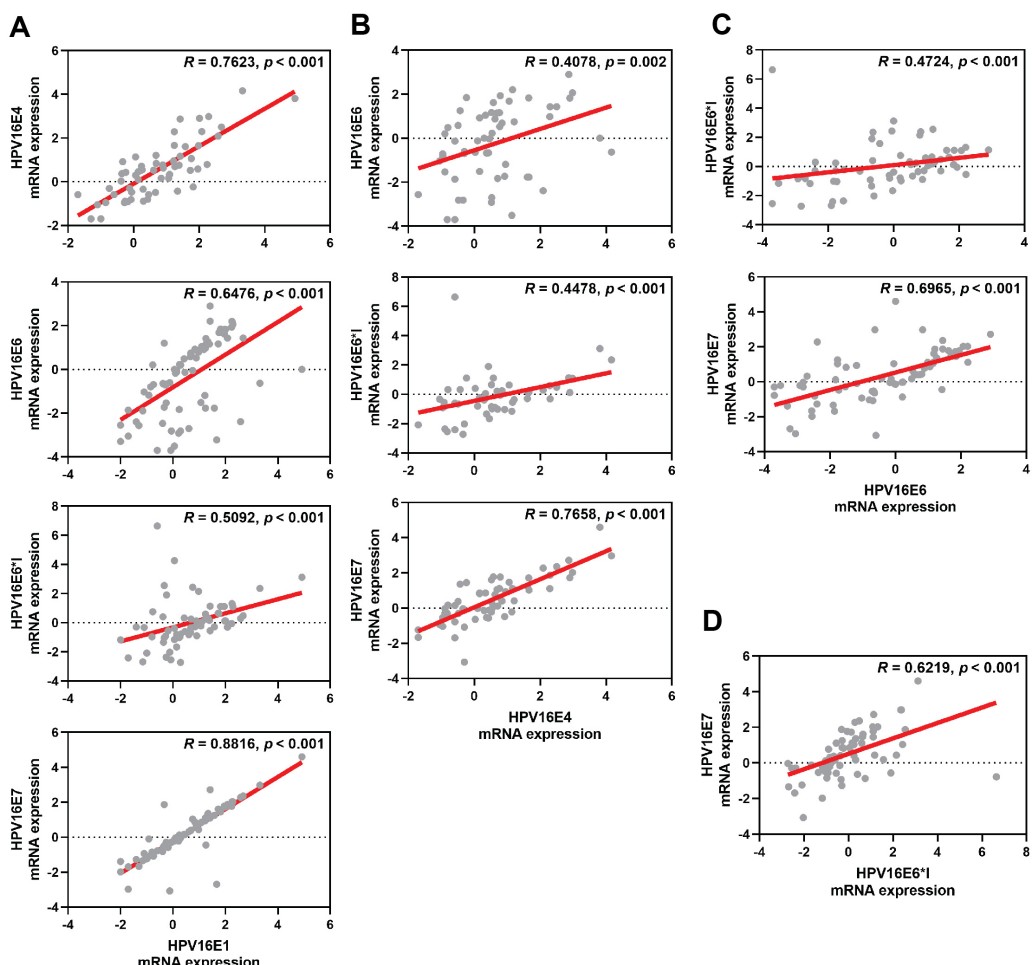

**Figure 2  Spearman's correlation among HPV16 transcripts.** (A) HPV16E1, (B) HPV16E4, (C) HPV16E6, and (D) HPV16E6*I.

ovarian serous cystadenocarcinoma (OV), and skin cutaneous melanoma (SKCM) was also demonstrated (Fig. S1).

The correlation between FOXO3a and those HPV early transcript expressions was also analyzed. No correlation between HPV16E1, E4, E6, E6*I, E7, and FOXO3a expression was found (Fig. 4). However, FOXO3a was downregulated in the higher stage of the disease (Fig. 3). To explore the expression level of FOXO3a along with HPV-related cervical carcinogenesis, the correlation between FOXO3a expression and the target of HPV16E6 (p53) and HPV16E7 (pRB and p107) oncoproteins was observed through GEPIA2 using TGCA (CESC Tumor/Normal) as a database. The results showed the positive correlation between FOXO3a expression and all genes (p53, pRB, and p107) (Figs. S2 and S3).

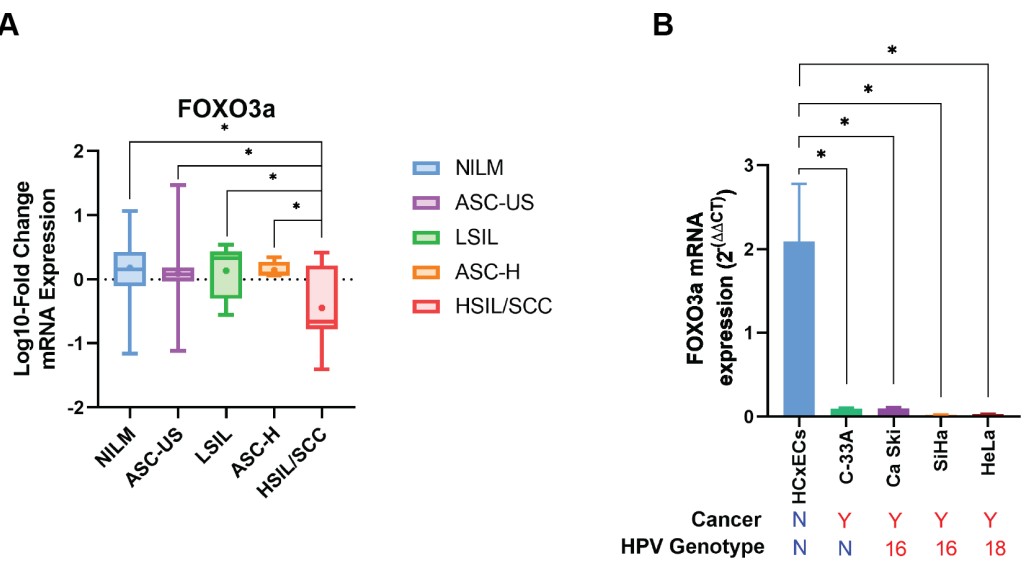

**Figure 3 FOXO3a mRNA expression.** (A) Fold-change expression of FOXO3a at in different cytology stages. The box plot showed median (line) and mean (dot) of log10-fold change mRNA expression. (B) FOXO3a mRNA expression in primary cervical epithelial cells (HCxECs) and cervical cancer cell lines. Bar chart represents the fold change of FOXO3a expression (Mean ± SEM).

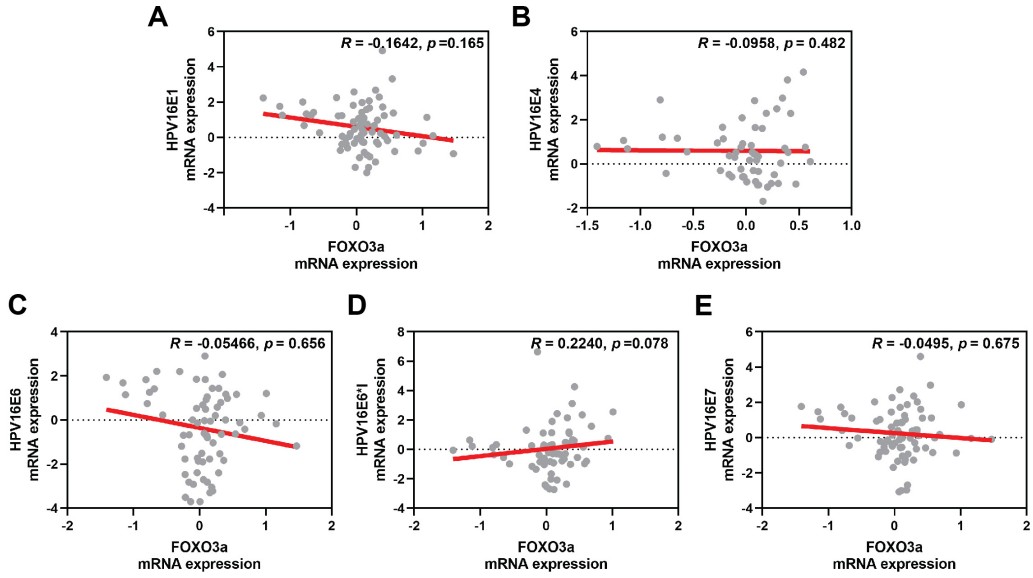

**Figure 4 Spearman's correlation between FOXO3a expression and HPV16 transcripts.** (A) HPV16E1, (B) HPVE4, (C) HPV16E6, (D) HPV16E6*I, (E) HPV16E7.

# DISCUSSION

HPV infection is the primary risk factor for the development of cervical cancer. E5, E6, and E7 expressed by hrHPVs are known to enhance the development of cancer (*Baedyananda*

*et al., 2022*). Nevertheless, the role of other early HPV genes in association with viral carcinogenesis is still unclear. Hence, in this study, the E1, E4, E6, E6*I, and E7 genes were selected. Only a limited number of studies have investigated HPV16E1 expression in relation to cervical cancer development. A study by *Baedyananda, Chaiwongkot & Bhattarakosol (2017)* revealed that HPV16E1 expression was related to the severity of the disease. Among HPV16-positive cervical swab specimens characterized by histology results, HPV16E1 expression in the CIN2/3 and squamous cell carcinoma (SCC) samples was significantly higher than that in normal samples (*Baedyananda, Chaiwongkot & Bhattarakosol, 2017*). This observation was also confirmed in this study, although cytological characteristics were used (Fig. 1A). HPVE1 is known to function as a helicase, which is able to cause DNA damage; however, dysregulated cellular genes have been observed in HPV16 E1 overexpressing cells (*Baedyananda et al., 2022*). Several host genes involved in immune response (ISG20), metabolism (ALDOC), protein synthesis (RPL36A), DNA damage (ATR, BRCA1, and CHK1), and cell proliferation (CREB5, HIF1A, NFKB1, PIK3CA, JMJDIC, TSC22D3, and FOXO3) have been shown to be significantly downregulated when HPV16E1 were upregulated (*Baedyananda et al., 2021*). The expression of the genes involved in the toll-like receptor, interferon, and apoptosis pathways, as well as the antiviral interferon-stimulated gene set, was also modulated by HPV18 E1 (*Castillo et al., 2014*). Moreover, the helicase function itself can also play some role in the development of cancer. Genome instability, a hallmark of cancer (*Hanahan & Weinberg, 2011*), results from various factors that prompt damage to DNA (*Langie et al., 2015*). This damage can be worsened by impaired DNA repair mechanisms and malfunctioning tumor suppressor molecules (*Negrini, Gorgoulis & Halazonetis, 2010*). It is possible that the host genome can experience double-strand DNA breaks due to the action of HPV E1 proteins. To better understand HPV16 E1's role in carcinogenesis, more research should be done on the virus's capacity to directly damage host DNA and the consequences of carcinogenesis. Recently, HPV16E1 expression was shown to be involved in several characteristics of cancer cells, including cell viability, anchorage-independent cell growth, and cell invasion properties, suggesting the possible role of HPV16E1 as an oncoprotein (*Sasivimolrattana, Chaiwongkot & Bhattarakosol, 2023*). In addition, it was also found that HPV16E7 protein was downregulated in HPV16E1 knocked-down cells (*Sasivimolrattana, Chaiwongkot & Bhattarakosol, 2023*). This suggested that there might be some correlation between E1 and E7. Here, HPV16E7 mRNA expression was shown to be significantly increased (Fig. 1E), similar to HPV16E1, with well correlation to HPV16E1 expression ($R = 0.8816$, $p < 0.001$, Fig. 2). E7 is known to interact with the pRB to regulate the growth of cells (*Egawa & Doorbar, 2017*). Increased expression of the E7 oncoprotein in cervical cancer is correlated with a lower level of pRB proteins due to a high viral load (*Mir et al., 2023*).

Interestingly, HPV16E6 and E6*I expression had no significant difference among various pre-cancerous stages (Figs. 1C and 1D). The HPV16E6 oncoprotein interacts with p53 to block apoptosis, leading to cell transformation (*Egawa & Doorbar, 2017*). Previous studies demonstrated that HPV16E6 expression is upregulated by disease severity (*Chang et al., 2021*; *Tagle et al., 2014*). However, in our study, no significant difference between HPV16E6 expression and disease progression was found. We presume that this might be due to the

moderate number of samples in our study. If the number of samples is increased, the differences might be observed since a trend towards an increase in HPV16E6 was found in the HSIL/SCC group when compared to others (Fig. 1C). In the case of E6*I, the expression was stable in all groups (Fig. 1D). This result contradicted the previous study showing that in HSIL/cervical cancer samples, HPVE6*I mRNA expression was higher than in LSIL samples (*Schmitt et al., 2011*). While increasing the mRNA gap between the stop codon of E6 and the start codon of E7 to facilitate better ribosome assembly is one of the most well-known functions of the E6*I transcript (*Zheng et al., 2004*). In contrast, some research suggested that E6*I might have an opposite function, such as anti-tumor properties (*Guccione, Pim & Banks, 2004*; *Pim, Massimi & Banks, 1997*). As a result, the expression and functions of E6*I in cervical carcinogenesis are still unclear. Additional functional studies on the potential roles and mechanisms of E6*I in cervical carcinogenesis should be warranted. Taken together, according to our findings, E6*I might not be involved in the progression of cervical carcinogenesis.

HPV E4 is one of the HPV genes that is expressed when the virus is in the late life cycle during infection. It has been proposed that this expression aids in the amplification of viral DNA and the release of virion (*Doorbar, 2013*). E4 expression is predominant in the fully differentiated keratinocytes' upper layer, suggesting that it is associated with active infection (*Doorbar, 2013*; *McBride, 2022*). In this study, E4 expression peaked in LSIL and differed significantly from NILM. Furthermore, compared to NILM, there was a notable increase in E4 expression in ASC-US (Fig. 1B). After the LSIL stage, E4 expression was stably expressed. These might indicate that E4 was expressed during a productive infection, which was identified by the presence of koilocytes, an indication of LSIL (*Griffin et al., 2020*). Moreover, high-grade lesions with low E4 expression exhibited a correlation with increased hypermethylation, a marker for HPV-induced cervical cancerous lesions that indicates the development of cervical cancer (*Zummeren et al., 2018*). In addition, the E2/E4 genomic region showed an increase in DNA hypermethylation with higher disease severity. Samples with high-grade lesions (CIN3) had a methylation level twice as high as those with persistent infections (CIN1, more than two years of HPV16 infection), where the methylation suppressed the expression of the E2/E4 gene (*Mirabello et al., 2012*).

The pro-apoptotic and antiproliferative properties of FOXO3a have led to the suggestion that it functions as a tumor suppressor in a variety of cancers (*Farhan et al., 2017*). A previous study from our group revealed that the expression of P-FOXO3a, an inactive form of FOXO3a, was altered in HPV16E1 knocked-down cells (*Sasivimolrattana, Chaiwongkot & Bhattarakosol, 2023*). To enhance comprehension of the relationship between FOXO3a expression and the advancement of disease, FOXO3a mRNA expression in cervical precancerous samples was observed. There was a decrease in FOXO3a mRNA expression in the HSIL/SCC group when compared to other lower groups (Fig. 3A). In addition, when comparing the expression of FOXO3a mRNA in the primary cervical epithelial cells (HCxECs) to several kinds of cervical cancer cell lines with HPV infection (HPV16-positive CaSki and SiHa cells; HPV18-positive HeLa cells) and without HPV infection (C-33A cells), regardless of whether the cells had an HPV infection or not, all cervical cancer cells had downregulated expression of FOXO3a mRNA. This suggested that a common feature of

cervical cancer cells is the downregulation of FOXO3a. The downregulation of FOXO3a in cervical cancer samples was also supported by the meta-analysis using GEPIA2 (Fig. S1). A similar pattern of FOXO3a downregulation was also found in other kinds of cancer, such as ovarian cancer. To better know the relation about the attenuation of FOXO3a expression and HPV gene expression, the correlation between FOXO3a mRNA expression and p53, pRB, and p107, which are the well-known targets of HPV16 oncoproteins, was observed. As predicted, the positive correlation was found, suggesting that in the progression of HPV carcinogenesis, FOXO3a was downregulated along with other genes. Note that FOXO3a is also downregulated in C-33A cells that are HPV-negative (Fig. 3B), suggesting that the downregulation of this gene might be a property of cervical cancer cells. At present, no evidence of a protein-protein interaction between the HPV16 oncoproteins and FOXO3a was shown. Additional studies are needed to fully understand their interaction. Interestingly, our findings aligned with a prior study about another isoform of FOXO that noted reduced FOXO1 mRNA expression in both primary tumor biopsies and cell lines of cervical cancer (*Prasad et al., 2014*). Furthermore, these patterns of FOXO downregulation were also observed in other cancer types, such as non-small cell lung cancer (*Maekawa et al., 2009*), endometrioid endometrial carcinoma, and non-endometrioid cancers (*Risinger et al., 2003*). Thus, there may be a negative correlation between the severity of the disease and FOXO3a mRNA expression. According to our results, it was noted that the downregulation of FOXO3a might not be specifically related to HPV infection. In addition, to ensure the potential of FOXO3a to be a novel biomarker for cancer progression, a larger number of samples should be included. A correlation between histological characteristics, including CIN1-3 and cancer, and FOXO3a expression should be observed.

## CONCLUSIONS

This study sheds light on HPV16E1, showing that elevated expression of HPV16E1 correlates with increased disease severity, suggesting HPV16E1 might be an oncoprotein similar to other HPV oncoproteins (E5, E6, and E7). Targeting these viral genes could become a focus for the development of antiviral or anticancer treatments specifically tailored to disrupt these pathways and halt progression from low-grade to high-grade lesions. Notably, HPVE4 mRNA expression was consistently detected in all LSIL samples, contrasting with other HPV early transcripts. In addition, a significant increase in HPV16E4 mRNA expression was observed from NILM to ASCUS and LSIL. In contrast to HPV gene expression, human FOXO3a mRNA expression decreased with disease severity. Hence, monitoring FOXO3a expression together with HPVE1 and E7 might serve as a prognostic marker for disease progression. In addition, restoring FOXO3a expression or enhancing its activity may serve as a therapeutic strategy for cervical cancer treatment. As a result, future studies about FOXO3a as a therapeutic strategy are warranted. Note that, in this study, the number of samples in some groups is low, which might be a limitation of this study.

## ACKNOWLEDGEMENTS

We are grateful to the Virology Unit, Department of Microbiology, Faculty of Medicine, Chulalongkorn University, for providing the leftover specimens of the HPV16-positive cervical swab samples. In addition, we gratefully acknowledge Supranee Buranapraditkun, Ph.D., Center of Excellence in Vaccine Research and Development (Chula VRC), Faculty of Medicine, Chulalongkorn University, Bangkok, Thailand, for supporting the culture of primary cells.

### Funding

This work was supported by the Chulalongkorn University (Fundamental Fund 66, grant number HEA663000053 by Parvapan Bhattarakosol). The funders had no role in study design, data collection and analysis, decision to publish, or preparation of the manuscript.

### Grant Disclosures

The following grant information was disclosed by the authors:
The Chulalongkorn University: HEA663000053.

### Competing Interests

The authors declare there are no competing interests.

### Author Contributions

- Thanayod Sasivimolrattana conceived and designed the experiments, performed the experiments, analyzed the data, prepared figures and/or tables, authored or reviewed drafts of the article, and approved the final draft.
- Aileen Gunawan performed the experiments, analyzed the data, prepared figures and/or tables, authored or reviewed drafts of the article, and approved the final draft.
- Warattaya Wattanathavorn performed the experiments, prepared figures and/or tables, and approved the final draft.
- Chavis Pholpong performed the experiments, prepared figures and/or tables, and approved the final draft.
- Arkom Chaiwongkot analyzed the data, authored or reviewed drafts of the article, and approved the final draft.
- Pattarasinee Bhattarakosol analyzed the data, authored or reviewed drafts of the article, and approved the final draft.
- Parvapan Bhattarakosol conceived and designed the experiments, analyzed the data, authored or reviewed drafts of the article, supervision, funding acquisition, project administration, and approved the final draft.

### Human Ethics

The following information was supplied relating to ethical approvals (*i.e.*, approving body and any reference numbers):

The study was conducted in accordance with the Declaration of Helsinki and received ethical approval from the Institutional Review Board of the Faculty of Medicine, Chulalongkorn University, Bangkok, Thailand on January 18, 2023 (IRB No. 0035/66).

## Ethics

The following information was supplied relating to ethical approvals (*i.e.*, approving body and any reference numbers):

The study was conducted in accordance with the Declaration of Helsinki and received ethical approval from the Institutional Review Board of the Faculty of Medicine, Chulalongkorn University, Bangkok, Thailand on January 18, 2023 (COA No.0051/2023, IRB No.0035/66). The protocol of the study was also approved by Institutional Biosafety Committee (IBC) of the Faculty of Medicine, Chulalongkorn University (MDCU-IBC006/2023).

## Data Availability

The raw data are available in the Supplemental Files.

## Supplemental Information

Supplemental information for this article can be found online at http://dx.doi.org/10.7717/peerj.18601#supplemental-information.

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
