# Peer review of "Upregulation of HPV16E1 and E7 expression and FOXO3a mRNA downregulation in high-grade cervical neoplasia"

_PeerJ, doi:10.7717/peerj.18601_

## Round 0.1 · original submission · Major Revisions

Please address concerns of both reviewers and amens manuscript accordingly.

Although the comments from R1 are brief, they are important and you should clearly address the concerns which they are indicative of. To give extra context to their comments, they believe that the LSIL, ASC-H, and HSIL/SCC group numbers are too small to make any conclusions. And they feel that the conclusions that this data provides valuable insights into molecular mechanism is an overstatement based on the current data presented.

Reviewer 1 ·

Basic reporting

All basic reporting elements met.

Experimental design

Some experimental groups too small and HSIL and SCC should not be combined. Not clear why cell lines were included in this study

Validity of the findings

Hard to interpret findings given small sample sizes. Conclusions overstated.

Reviewer 2 ·

Basic reporting

The writing is clear and easy to read. The introduction successfully sets up the hypothesis that HPV E1 expression will be inversely correlated with FOXO3a. Citations are relevant but fairly sparse, with a high reliance on review articles. This aspect to basic reporting could be improved in ways that better guide the reader to easily identify the most rigorous background information. Figures are easy to follow but some experimental details are lacking regarding time points, etc. in the results and figured legends (Figure 2 and 4). Raw data is provided.

Experimental design

This is a straightforward study that performs qRT-PCR on cervical swab samples from cancer patients with the goal of tracking changes to HPV E1 and host FOXO3a expression during cancer progression. The methods are described well overall. Regarding experimental rigor, two comments. First, the specificity of the HPV primer probe sets is not rigorously assessed. Lines 129-131 state that specificity was addressed using a no template control (and references prior publications) but this sounds like inadequate comparator. Better would be comparisons of HPV+/- tissues / cell lines with the same primers used in the study like is done for FOXO3a in Figure 4. A second concern (noted by the authors) is the small size of the dataset. Project design could be strengthened by adding comparisons/metanalysis to consider other HPV-associated gene profiling studies, such as cross-referencing the TCGA database to expand their analysis and add more samples to their disease groups (late-stage disease groups here were particularly lacking).

Validity of the findings

While the findings overall may be valid, a concern with this manuscript is frequent overstatement. The data are correlations - any reference to causation (and there are many) is speculative and should be removed, starting with the title and extending to the end of the paper that claims “mechanistic insights”. There is limited evidence here that supports the hypothesis that E1 or E7 are oncogenic or that FOXO3a supports “..the progression of HPV16-induced cervical cancer”. Indeed, Figure 4B actually shows that FOXO3a is downregulated in HPV-negative C33A cells, a result that frowns on a cause-effect relationship with E1, etc. Moreover, based on Figure 4A any correlation between HPV gene expression and FOXO3a is weak except for during full blown cancer. I think that what you can conclude from these data are that HPV E mRNA levels including for E1, E4 and E7 increase during cancer progression, and that downregulation of FOXO3a mRNA is a characteristic of HPV cancer cells and HSIL/SCC. It is fine that the dataset is hypothesis-generating but needs to be treated as such. Alternatively, the authors could consider expanding the cell culture studies to overexpress and knockdown E1, E4 and E7 individually and in tandem and readout markers of FOXO3A gene expression.

---

## Round 0.2 · Minor Revisions

Please carefully address remaining issues pointed by the reviewer and revise manuscript accordingly.

Reviewer 2 ·

Basic reporting

Basic reporting requirements met.

Experimental design

In the revision, the authors claim that their primers are specific in the methods but these data are not included- need to be.

Validity of the findings

Comments are minor but should be addressed prior to publication:

1. Lines 231-238: How does a FOXO3a correlation with p53, pRB, p107 levels relate to a correlation with E6 and E7 levels? FOXO3a is downregulated in C33A cells that are HPV-negative, hence the correlation seems to be with cancer and not HPV. Needs to be more clear that this is case. I don't see much support for the hypothesis that there is a correlation between E1 levels and FOXO3a. More likely this study disproves the hypothesis.

2. Lines 1-3: 347: E7 is already known to be associated with disease progression. E1 may or may not be. Check sentence for accuracy.

3. In the revision, The authors have expanded their analysis by comparing their findings to a database. They suggest the new metaanalysis “confirms” experimental results (lines 37, 216, 325). "Confirms" is not accurate language. “Supports their conclusions”?

4. Title states that E1 and E7 are elevated and FOXO3 down in cancer.But based on Fig1 all HPV transcripts except for E6*I are elevated?

---

## Round 0.3 · accepted · Accept

All remaining issues indicated by the reviewer were addressed and revised manuscript is acceptable now.